# Heparanase Expression Propagates Liver Damage in CCL4-Induced Mouse Model

**DOI:** 10.3390/cells11132035

**Published:** 2022-06-27

**Authors:** Xiaowen Cheng, Juan Jia, Tianji Zhang, Xiao Zhang, Israel Vlodavsky, Jin-ping Li

**Affiliations:** 1SciLifeLab Uppsala, The Biomedical Center, Department of Medical Biochemistry and Microbiology, University of Uppsala, 75237 Uppsala, Sweden; chengxw34@hotmail.com (X.C.); juan.jia@regionuppsala.se (J.J.); 2Division of Chemistry and Analytical Science, National Institute of Metrology, Beijing 100029, China; zhangtianji@nim.ac.cn; 3Department of Neuroscience, University of Uppsala, 75237 Uppsala, Sweden; xiao.zhang@neuro.uu.se; 4Cancer and Vascular Biology Research Center Rappaport, Faculty of Medicine, Technion 31096, Israel; vlodavsk@mail.huji.ac.il

**Keywords:** heparanase, liver damage, fibrosis, autophagy

## Abstract

Heparanase is elevated in various pathological conditions, primarily cancer and inflammation. To investigate the significance and involvement of heparanase in liver fibrosis, we compared the susceptibility of wild-type (WT) and heparanase-overexpressing transgenic (Hpa-tg) mice to carbon tetrachloride (CCL4)-induced fibrosis. In comparison with WT mice, Hpa-tg mice displayed a severe degree of tissue damage and fibrosis, including higher necrotic tendency and intensified expression of smooth muscle actin. While damage to the WT liver started to recover after the acute phase, damage to the Hpa-tg liver was persistent. Recovery was attributed, in part, to heparanase-stimulated autophagic activity in response to CCL4, leading to increased apoptosis and necrosis. The total number of stellate cells was significantly higher in the Hpa-tg than the WT liver, likely contributing to the increased amounts of lipid droplets and smooth muscle actin. Our results support the notion that heparanase enhances inflammatory responses, and hence may serve as a target for the treatment of liver damage and fibrosis.

## 1. Introduction

Liver fibrosis is the pathologic result of chronic inflammatory liver diseases, characterized by the excessive accumulation of extracellular matrix (ECM) and involving activated Kupffer cells and hepatic stellate cells (HSC) [1]. Kupffer cells promote inflammatory and fibrogenic responses through the release of cytokines and chemokines that aggravate inflammation and participate in the activation and trans-differentiation of HSC [1]. This phenotypic change transforms HSC into cells that express alpha-smooth muscle actin (α-SMA), proliferate, and deposit ECM proteins that progressively accumulate in the liver [2]. Clinically, the main causes of liver fibrosis include chronic hepatitis B virus (HBV) infection, alcohol abuse, and nonalcoholic steatohepatitis (NASH) [3]. Advanced liver fibrosis results in cirrhosis, leading to hepatocellular dysfunction, irreversible liver failure, and cancer.

Heparan sulfate (HS) proteoglycans (HSPG) constitute a major component of the ECM, maintaining, among other multiple biological functions, the integrity of ECM [4]. Heparanase (Hpa) is the sole endoglucuronidase that specifically degrades HS in the ECM, and thereby modulates the ECM’s structure and bioavailability of HS-bound pro-inflammatory and pro-tumorigenic factors [5]. Alterations in HS and elevated levels of heparanase expression have been found in diseased liver, correlated with the degree of liver fibrosis and hepatocellular cancer [6,7], implying that heparanase plays a role in the pathogenesis of liver diseases. We have previously reported that overexpression of heparanase in mouse liver led to the production of structurally altered HS, characterized by shorter chains and a substantially higher degree of sulfation, as well as rapid turnover and high affinity to FGF2 and its receptors [8]. Considering the multiple roles of HSPG in the ECM and cell surface, we hypothesized that heparanase degradation of HS affects the functions of HS under conditions of liver injuries. The present study is aimed at elucidating the impact of heparanase on the pathogenesis of liver fibrosis, using the well-established CCL4-induced mouse model. Briefly, WT and heparanase-overexpressing (Hpa-tg) mice were treated with one or two peritoneal injections of CCL4 and sacrificed on days 2 and 7. Histological and biochemical analyses revealed considerably severe pathological alterations in Hpa-tg vs. WT mice, ascribed, in part, to heparanase overexpression.

## 2. Materials and Methods

### 2.1. Mice

The mouse strains used in this study were Hpa-tg mice overexpressing human heparanase [9] in C57BL/6 and Balb/c genetic backgrounds. Wild-type C57BL/6 and Balb/c mice were used as controls. Mice were bred and maintained in the animal facility located at the Biomedical Center, Uppsala University. The local ethics committee (Uppsala djurförsksetiska nämnd) approved the procedures (01751/2020) involving the animal experiments, and the study was conducted in accordance with animal welfare regulations.

### 2.2. CCL4 Induced Liver Damage

Adult mice were injected (days 1 and 4) intraperitoneally with CCL4 (Merck, Rahway, NJ, USA) mixed with olive oil (1:3) at a dose of 0.5–1.0 µL of CCL4/g body weight. The animals were monitored 2–3 times daily. Upon termination, the mice were sacrificed and blood was collected by cardiac punctuation, followed by perfusion with 30 mL of PBS. The livers were dissected and divided into two parts; one part was immediately frozen in dry ice for biochemical analyses, and the other part was fixed in paraformaldehyde for histological studies [10].

### 2.3. Analysis of Plasma

Blood was collected in EDTA anticoagulation tubes. After centrifugation, plasma was collected for analysis of p-ASAT (aspartataminotransferase) and p-ALAT (alaninaminotransferase), carried out at the clinical biochemical laboratory of the University Hospital, Uppsala, Sweden.

### 2.4. Morphological and Histological Analyses

Tissue specimens were fixed and embedded in paraffin. Sections (4 µm) of the fixed tissues were histologically or immunologically stained as indicated in the figure legends. The antibodies used are listed in Table 1. All images were captured by microscopy (Nikon 90i, Nikon, Tokyo, Japan), and quantification of the positive signals was performed by Image J software.

### 2.5. Tunel Assay

Paraffin-embedded tissue sections were stained with the in situ cell death detection kit (Roche, 11684795910, Roche, Basel, Switzerland) according to the manufacturer’s instructions. At least 5 fields of each section were randomly selected, and the number of positive cells in each field was counted using Image J software.

### 2.6. Western Blotting Analysis

Frozen livers were homogenized in RIPA buffer containing protease inhibitors and incubated on ice for 30 min. After centrifugation, the supernatant was collected and protein concentration was determined with the BCA method. Proteins (40 µg of cell lysates) were separated by gel electrophoresis (SDS-PAGE) and electroblotted onto PVDF membrane. The membranes were blocked with 5% BSA or milk for 1 h at room temperature and then probed by primary antibodies (shown in Table 1) overnight at 4 °C. After washing ×3 in TBST buffer, membranes were incubated with secondary antibodies for 1 h at room temperature. Signals were detected by ECL (Chemidoc mp imaging system, Bio-Rad, Hercules, CA, USA) and quantified using Image Lab software (Bio-Rad Laboratories).

### 2.7. Isolation and Analysis of Hepatocytes

The two-step collagenase digestion method [11] was used with slight modifications. In brief, livers were perfused by HBSS buffer containing 5 mM glucose, 0.5 mM EGTA, and 25 mM HEPES (pH 7.4 at 37 °C), followed by perfusion with DMEM containing 15 mM HEPES and collagenase (Type IV, Worthington, 100 CDU/mL). The livers were gently removed from the culture dish and suspended in DMEM containing 10% FBS. The digested tissue was passed through a 70 µm nylon strainer, then centrifuged at 50× *g* for 2 min. The cell pellet was suspended in DMEM supplemented with Pen/Strep, 15 mM HEPES, 100 nM dexamethasone, and 10% FBS and cultured on collagen I coated dishes. After 2 h, the non-attached cells were removed and the attached cells were cultured in fresh medium. For analysis of autophagy activity, cells were transferred to DMEM without FBS, and 0.5 × 10^6^ cells were seeded into a 6-well plate and cultured overnight. Then, chloroquine (CQ) was added to each well to a final concentration of 20 µM, and the cells were cultured for different periods as indicated in the figure legends. For Western blot analysis, cells were collected and lysed in lysis buffer (0.5% Triton X-100, 0.5% sodium deoxycholate, 20 mM Tris pH 7.4, 150 mM NaCl, 10 mM EDTA) supplemented with protease inhibitor cocktail (Thermo Scientific, Waltham, MA, USA). The cell lysates were analyzed as described above.

### 2.8. Isolation and Characterization of Hepatic Stellate Cells (HSC)

The procedure is essentially as described [12]. In brief, the liver was perfused in vivo with HBSS buffer containing pronase (Sigma, P5147, Sigma-Aldrich, Saint Louis, MS, USA) and collagenase D (about 20 mL with a flow rate of 2–3 mL/min) via the portal vein, followed by ex vivo digestion in HBSS buffer containing pronase, collagenase D, and DNase. The HSC were isolated by gradient centrifugation in Histodenz (Sigma, D2158). The layer containing stellate cells was collected and washed twice with HBSS by centrifugation. The cells were resuspended in DMEM containing 10% FBS and seeded in 6-well plates for analysis.

## 3. Results

### 3.1. Enhanced Damage of Hpa-Tg Liver in Response to CCL4

Hpa-tg and WT mice were treated with CCL4 mixed (1:3) with olive oil (intraperitoneal injection of 0.5–1.0 µL/g body weight on day 1 or days 1 and 4). Upon sacrificing the mice on days 2 and 7, plasma and livers were collected for biochemical and histological analyses. Clinical lab analyses revealed elevated levels of both ALAT (alaninaminotransferase) and ASAT (aspartataminotransferase) in the plasma of mice 7 days after CCL4 induction in both groups (Figure 1). The level of the enzymes was moderately increased in WT but markedly elevated in the Hpa-tg livers. Notably, naive Hpa-tg plasma contained significantly higher basal levels of both ALAT and ASAT compared with naïve WT plasma. Next, we examined the necrotic conditions of sections from paraffin-embedded liver tissues. Staining with anti-heparanase antibody confirmed overexpression of heparanase in the Hpa-tg liver, but its expression level did not show an obvious change upon CCL4 induction (Appendix A). The endogenous heparanase in naïve WT liver was not detectable, nor in WT livers collected 2 days post-CCL4 stimulation. Heparanase was slightly elevated in WT livers 7 days post-stimulation (Appendix A). H&E staining revealed large necrotic areas in livers collected 2 days after CCL4 stimulation (one injection) to a similar degree in both the WT and Hpa-tg livers (Figure 2). The damaged areas were substantially reduced in sections of WT liver 7 days after CCL4 stimulation, but not in the Hpa-tg mice, indicating a slower wound healing process in the Hpa-tg liver. This result was further confirmed by the Tunel assay. Massive apoptotic cells were detected in the livers of both groups 2 days post-CCL4 stimulation, with a significantly higher level in the Hpa-tg liver (Figure 3). The number of apoptotic cells was almost non-detectable in WT liver 7 days post-stimulation, in line with the H&E staining, implying a recovery process. In comparison, the Hpa-tg liver still contained a large number of apoptotic cells, demonstrating a slow recovery process.

### 3.2. Higher Levels of Alpha-SMA in Hpa-Tg Liver

Immunostaining of liver sections detected collagen III-positive signals along the vessel walls in livers of both naïve WT and Hpa-tg mice. This staining was significantly intensified and expanded to smaller vessels from day 2 to day 7 after CCL4 induction, displaying a similar pattern in both groups (Figure 4). In contrast, alpha-smooth muscle actin (alpha-SMA) signals were non-detectable in the liver of naïve mice in both groups. Positive signals appeared 2 days post-stimulation in both the WT and Hpa-tg groups and became considerably stronger in livers collected 7 days post-stimulation (Figure 5). Quantification of the signals showed a significantly larger SMA-positive area in Hpa-tg vs. WT sections on day 7, showing a typical pattern of liver fibrosis along the sinusoids. Neither the collagen III nor the SMA signals overlapped with heparanase (HPSE) (Figure 4 and Figure 5). Western blot analysis of liver tissue lysates showed that MMP9 was significantly increased on day 2 after CCL4 induction, followed by reduced levels on day 7, with no difference between the WT and Hpa-tg mice (Appendix A).

### 3.3. Higher Autophagic Activity of Hpa-Tg Hepatocytes in Response to CCL4

Having seen a higher number of apoptotic cells in the Hpa-tg liver upon CCL4 induction, we examined markers for autophagy activity. Western blot analysis of liver tissue lysates revealed a much higher level of LC3-II in the Hpa-tg liver on day 2 (Figure 6), in accordance with the significantly higher number of apoptotic cells detected in the same liver samples (Figure 3). The LC3-II levels were dramatically decreased on day 7 in both groups, in line with the reduced apoptosis tendency (Figure 3), most likely reflecting a protective response. P62 was slightly increased in the Hpa-tg liver on day 2, followed by a marked decrease on day 7. Examination of additional autophagy-related proteins revealed substantially reduced levels of Atg3 and Atg 7, with no difference between the two groups (Appendix A). To further elucidate the effect of heparanase on autophagy, we analyzed hepatocytes isolated from WT and Hpa-tg naïve mice. Hepatocytes isolated from in vivo perfused livers did not show any morphological difference between Hpa-tg and WT mice (Appendix A). Western blot analysis showed slightly higher basal levels of LC3-II in Hpa-tg than in WT cells (Figure 7A; Appendix A). Upon chloroquine (CQ) treatment, LC3-II was steadily increased with time in both Hpa-tg and WT cells and reached a much higher level after 24 h of stress, most notably in the Hpa-tg hepatocytes (Figure 7A). Similarly, p62 was increased upon 24 h of CQ stress (Figure 7B). Immunostaining of the hepatocytes showed merging of the heparanase and LC3 signals in the Hpa-tg hepatocytes (Figure 7C). There were no obvious differences between the WT and Hpa-tg hepatocytes in other markers related to autophagy (Appendix A).

### 3.4. Hepatic Stellate Cells

Given that hepatic stellate cells play central roles in liver fibrosis [13], the intensified deposition of SMA in Hpa-tg liver upon CCL4 stimulation suggests a higher activity of hepatic stellates. To examine the phenotype, stellates were isolated from livers of naïve WT and Hpa-tg mice and cultured in DMEM containing 10% FBS. Quiescent cells gradually differentiated into myofibroblasts, with no obvious morphological differences between WT and Hpa-tg cells for the first 10 days in culture. Differences were noted after 2 weeks of differentiation reflected by much stronger immunostaining of SMA in the Hpa-tg cells (Figure 8A). We also observed a higher number of oil droplets in the Hpa-tg than the WT stellates, which decreased with differentiation of the cells (Figure 8B,C). Notably, we repeatedly obtained significantly more stellate cells derived from Hpa-tg vs. WT livers (Figure 8D), but the Hpa-tg HSC appeared smaller in size (Figure 8E).

## 4. Discussion and Conclusions

Liver fibrosis is one of the high prevalence diseases affecting about 5% of the general population, and 18–27% of the population has risk factors, which is increased rapidly with aging [3,14]. Among the animal models that were developed to understand the underlying molecular mechanisms of liver injury and fibrosis, CCL4-induced liver fibrosis is commonly used [15]. However, unlike the slow progression of the human disease, CCL4 induces robust and rapid liver fibrosis. Elevated levels of heparanase (HPSE) were previously reported in response to CCL4-induced liver fibrosis, declining during progression of the disease [16]. Likewise, in a model of thioacetamide-induced hepatic fibrosis in rats, heparanase protein levels were increased [17,18]. Several studies have reported upregulation of heparanase by tumor cells in liver biopsies derived from patients with hepatocellular carcinoma [19,20]. In contrast, the involvement of heparanase in chronic non-cancerous liver disease is poorly demonstrated, and controversial. Previous studies suggest the involvement of heparanase in the early stages of liver damage and indicate inflammatory macrophages as the main source of heparanase, mediating the activation/transition of hepatic stellate cells (HSC) into myofibroblasts [16,21]. Most interestingly, a recent study reported that inhibition of heparanase prevents liver steatosis in mouse models [22].

To further elucidate the contribution of heparanase, we applied WT and transgenic mice overexpressing heparanase (Hpa-tg mice) in the CCL4-induced liver damage model. We found a highly significant ~15-fold increase in the levels of AST and ALT in the plasma of Hpa-tg vs. WT mice, indicating more severe liver damage and dysfunction in the Hpa-tg mice. Since the basal levels of ASAT and ALAT were slightly higher in the Hpa-tg plasma, it is conceivable that the high heparanase activity caused minor damage to the liver of CCL4-untreated naïve Hpa-tg mice. This may explain our observation that newborns of Hpa-tg mice are on average smaller in size and need to be fed with soft food and require delayed weaning (unpublished results), which may be caused by a mild liver function impairment. Due to the high stringency of animal welfare regulation at our facility, we could not extend the experiment beyond 7 days and hence may have not detected the full amplitude of the heparanase pro-fibrotic effect. Moreover, because the Hpa-tg liver expresses high levels of heparanase [8], we could not detect further up-regulation of the enzyme in response to CCL4 stimulation. HE staining revealed similar damage to the liver 2 days after induction. However, while WT mice recovered on day 7, there were no signs of recovery in the Hpa-tg liver, indicating that heparanase expression delayed the recovery process.

Hepatic stellate cells (HSCs) are pericytes located in the perisinusoidal space of the liver. Quiescent HSCs function as a storage depot for retinoids (vitamin A and retinyl ester) in oil droplets, while upon activation, the retinoid contents are rapidly lost [23]. HSCs are activated when the liver is damaged, producing extracellular matrix molecules (i.e., Col III) and smooth muscle actin, thereby propagating liver fibrosis [24]. We found that the total number of HSCs is significantly increased in naïve Hpa-tg livers, with a higher number of lipid droplets in each Hpa-tg stellate cell. This, together with the higher level of SMA, likely contributed to the more severe liver damage observed in the Hpa-tg mice upon CCL4 induction.

Autophagy plays a crucial role in hepatic homeostasis, and its deregulation is associated with liver diseases [25]. Studies have demonstrated that transformation of quiescent HSCs from the adipogenic to the myofibroblast profile, a hallmark in the propagation of hepatic fibrosis, is mediated through stress-induced autophagy [25,26]. We previously reported that heparanase promotes autophagy by suppressing, among other effects, mTORC1 activity [27,28,29]. Indeed, Hpa-tg hepatocytes exhibited a slightly higher basal level of LC3, which was considerably increased upon chloroquine stress as compared with WT hepatocytes. Similarly, LC3-II was slightly increased in WT livers but was significantly elevated in Hpa-tg livers, 2 days after CCL4 induction. These results suggest that heparanase enhances autophagic cell death upon CCL4 stimulation, evidenced by the high number of apoptotic cells in the Hpa-tg livers. The rapid decrease in LC3-II, 7 days post-CCL4 stimulation, may suggest a protective effect to avoid further death of cells and facilitate the healing process.

The pathology of liver damage and consequently fibrosis involves multiple factors. For instance, myeloperoxidase levels are elevated in CCL4-induced liver injury [30]. Cyclooxygenase 2 plays a critical role in chronic inflammation and was detected in inflammatory cells of damaged livers [31]. Notably, attenuation of liver fibrosis/cirrhosis was synergistically attenuated in response to a combined treatment with mesenchymal stem cells and induced bone marrow-derived macrophages [32]. Collectively, our study further elucidates the pathological effect of heparanase in CCL4-induced liver damage. The enhancing effect of heparanase is likely contributed to by an augmented autophagic activity in response to the injurious stimuli. Heparanase-inhibiting strategies could provide a new option for protection against liver injury and fibrosis.

## Figures and Tables

**Figure 1 cells-11-02035-f001:**
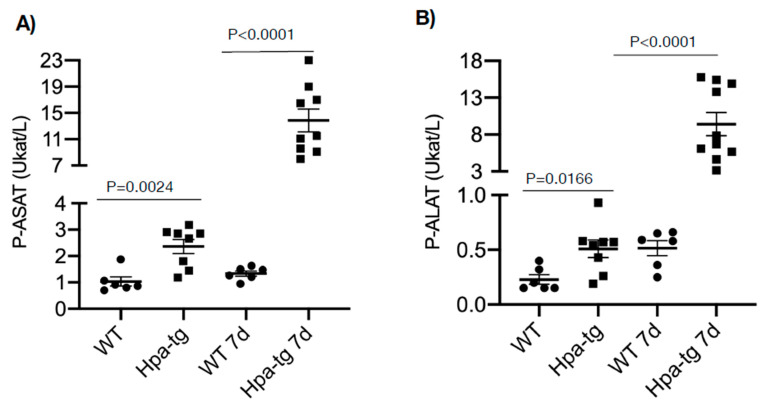
Plasma was collected from naïve and CCL4-treated mice and analyzed for p-ASAT (**A**) and p-ALAT (**B**) levels at the clinical biochemical laboratory, Akademiska hospital, Uppsala University. Higher levels of both enzymes were noted in naïve Hpa-tg vs. WT mice and even more so in CCL4-treated Hpa-tg vs. WT mice. (*n* = 6–10).

**Figure 2 cells-11-02035-f002:**
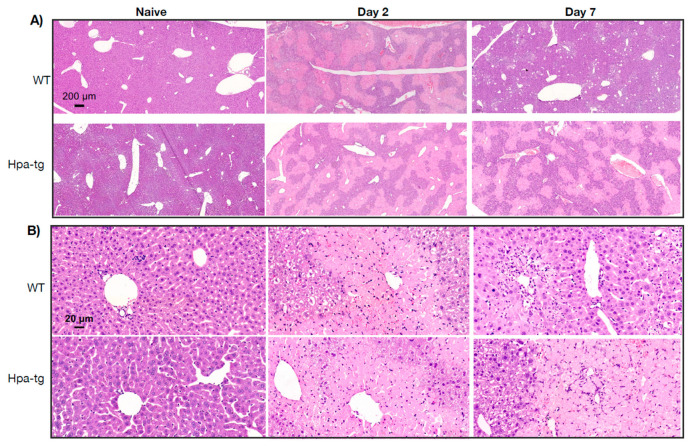
H&E staining of paraffin-embedded liver sections from naïve and CCL4 treated WT and Hpa-tg mice. (**A**,**B**) Large damaged areas are seen 2 days after treatment, with no substantial difference between the two groups. On day 7, the damaged area was markedly reduced in the WT but not the Hpa-tg liver. Shown are representative images at low (×5, upper panels) and high (×40, lower panels) magnifications.

**Figure 3 cells-11-02035-f003:**
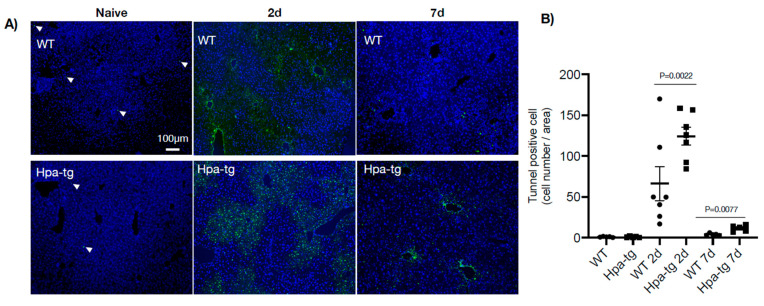
Cell apoptosis in livers from naïve and CCL4-treated mice. (**A**) Tissue sections derived from naïve and CCL4-treated WT and Hpa-tg livers were subjected to TUNEL staining. Representative images were counted for positively stained apoptotic cells (×10 magnification). Arrowheads: indicate trace TUNEL-positive cells in the naïve mice, set arbitrarily to a value of 1. (**B**) Average numbers of TUNEL-positive cells/microscopic field. *WT* and *Hpa-tg* represent naïve mice. The values are averages of more than 10 counted fields/mouse. (*n* = 4).

**Figure 4 cells-11-02035-f004:**
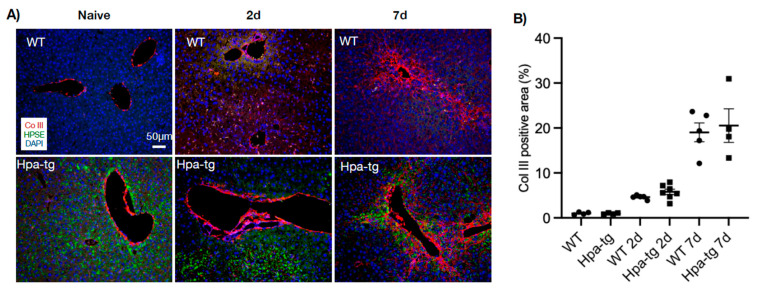
Immunohistostaining of collagen III (Col III, red) in paraffin-embedded liver sections. (**A**) In naïve mice, Col III was only detected along larger vessels. Upon CCL4 induction, the signals are intensified around veins and expanded to pericentral and periportal areas. (**B**) Col III -positive areas quantified by ImageJ. More than 10 random fields were quantified for each mouse (*n* = 4–7 per group). Staining intensity measured in WT naïve group was set arbitrarily to a value of 1. Heparanase (HPSE, green) was detected only in the Hpa-tg liver.

**Figure 5 cells-11-02035-f005:**
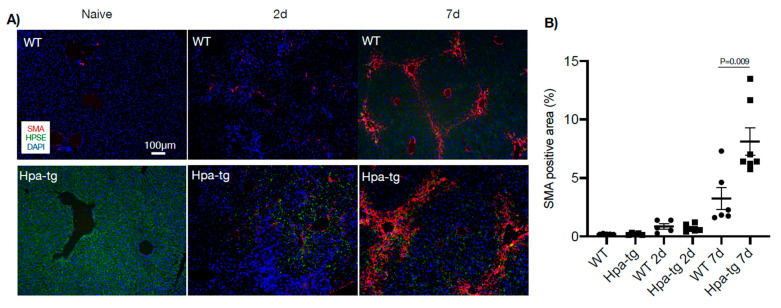
Immunohistostaining of alpha-smooth muscle actin (SMA, red) in paraffin-embedded liver sections. (**A**) SMA was non-detectable in naïve mice and a trace amount was detected 2 days after CCL4 induction. (**B**) Quantification of the positive areas by ImageJ revealed larger positive areas in Hpa-tg vs. WT liver. More than 10 random fields were quantified for each mouse (*n* = 4 per group), and the WT naïve value was set arbitrarily to a value of 1. Heparanase (HPSE, green) was detected only in the Hpa-tg liver.

**Figure 6 cells-11-02035-f006:**
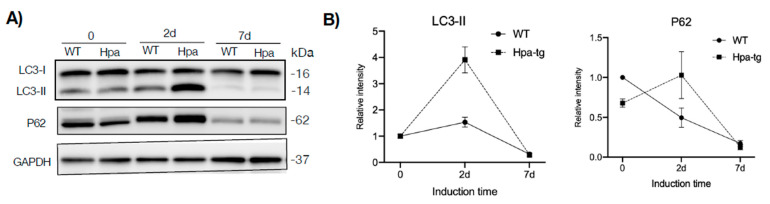
Autophagy. Frozen liver tissues were homogenized in RIPA buffer and total protein was quantified. (**A**) Equal amounts of tissue lysates (40 µg protein; *n* = 4–7 mice per group) were applied onto 8–15% SDS-PAGE and subjected to Western blotting probed with the indicated antibodies. (**B**) Band intensity was quantified by Image Lab software and the naïve WT value was set arbitrarily to a value of 1. Data are expressed as mean ± SEM of three experiments (each representing pooled liver tissue extracts derived from 4–7 mice per group).

**Figure 7 cells-11-02035-f007:**
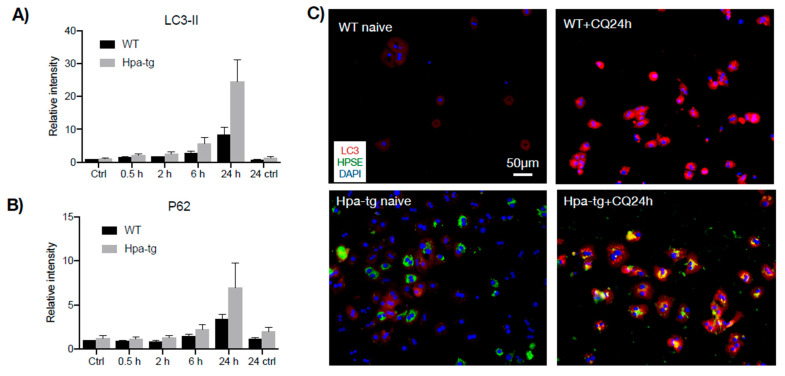
Autophagy in response to chloroquine. Hepatocytes isolated from naïve mouse liver were treated with chloroquine (CQ) to a final concentration of 20 μM and the cell lysates were subjected to Western blot analysis. (**A**) Quantification of LC3-II and (**B**) P62 levels (Western blot, Appendix A) in cell lysates prepared at the indicated time points (20 µg total protein/lane). Band intensity was quantified by Image Lab software and the naïve WT value was set arbitrarily to a value of 1 (each bar represents the average band intensity; *n* = 5 mice per group). (**C**) Immunostaining of hepatocytes before and after CQ treatment. Note merged signals of LC3 and heparanase in the Hpa-tg cells.

**Figure 8 cells-11-02035-f008:**
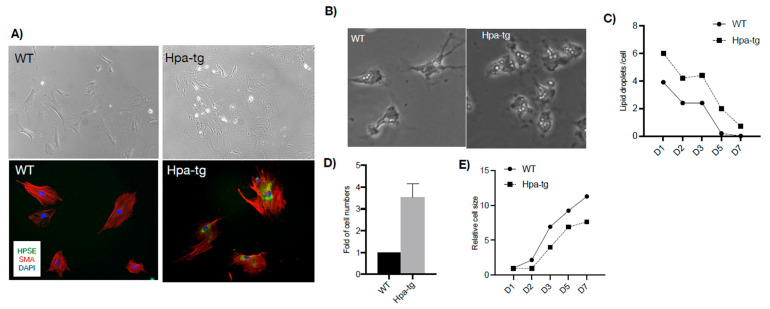
Stellate cells. (**A**) Phase contrast microscopy of cells cultured for 2 weeks (upper panels) and immunostained for alpha-smooth muscle actin (SMA, red) (lower panels), indicating differentiation of stellate cells to myofibroblasts. Heparanase (HPSE, green) was detected only in the Hpa-tg stellate cells. (**B**) Phase contrast microscopy and quantification (**C**) of oil droplets in WT and Hpa-tg stellate cells. Data are average numbers of at least 35 cells per group. (**D**) Relative number of stellate cells isolated from Hpa-tg and WT livers. Data are the average of 3 independent experiments (3 mice per group in each experiment). Naïve WT value was set arbitrarily to a value of 1. (**E**) Cell morphology was outlined as indicated in Appendix A using ImageJ, and the average cell area was measured and setting the value of naïve WT cells as 1. More than 35 cells per group were quantified.

**Table 1 cells-11-02035-t001:** Antibodies Used in this Study.

Antigen	Host	Working Dilution	Source and Code
Atg3	Rabbit	1:1000 (WB)	CST#3415
Atg7	Rabbit	1:1000 (WB)	CST#8558
Beclin-1	Rabbit	1:1000 (WB)	CST#3495
LC3I/II	Rabbit	1:1000 (WB)	CST#12741
P62	Rabbit	1:1000 (WB)	CST#23214
MMP9	Rabbit	1:1000 (WB)	Abcam#76003
GAPDH	Rabbit	1:2000 (WB)	CST#2118
Collagen III	Rabbit	1:100 (IHC)	Abcam #7778
HPSE	Mouse	1:200 (IHC)	In-house monoclonal
SMA	Rabbit	1:100 (IHC)	CST #19245

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
