# Peer review of "Heparanase Expression Propagates Liver Damage in CCL4-Induced Mouse Model"

_cells, 2022, doi:10.3390/cells11132035_

Round 1

Reviewer 1 Report

Dear authors,

the manuscript at hand concerning the role of heparanase in CCl4-induced liver damage is well written and clear. The topic is of interest as liver diseases are of growing importance particularly in the Western world.

However, there are several issues/clarifications that should be addressed.

1) Which imaging system was used for the detection of ECL for the western blot analyses?

2) The figure legend of figure 6 states that 'three representative experiments (4-7 mice/group)' are used for the graph. Were the lysates of the mice pooled and three technical replicates of the western blots conducted or were there 4-7 western blots per group and three "representative" ones were included for the graph. I would recommend to do individual western blots and include all data regardless that the number of replicates are different in the groups.

3) Figure legends 7 and 8 contain methodological explanations that are better suited in the materials and methods section.

4) There are 6 supplementary figures to the manuscript, but only two of them are referenced in the manuscript as suppl. figures 5 and 8 which doesn´t match the naming in the suuplementary files.

5) The following typos are used consistently throughout the manuscript:

'oliver oil" instead of 'olive oil'

'CCL4' instead of 'CCl4'

'imag lab' instead of 'image lab'

Best Regards!

Author Response

Reviewer 1

the manuscript at hand concerning the role of heparanase in CCl4-induced liver damage is well written and clear. The topic is of interest as liver diseases are of growing importance, particularly in the Western world.

However, there are several issues/clarifications that should be addressed.

  • Which imaging system was used for the detection of ECL for the western blot analyses?

Response: We have used Chemidoc mp imaging system (Bio-Rad). This information is now inserted in the 'Methods' section (p. 3, line 98).

  • The figure legend of figure 6 states that 'three representative experiments (4-7 mice/group)' are used for the graph. Were the lysates of the mice pooled and three technical replicates of the western blots conducted or were there 4-7 western blots per group and three "representative" ones were included for the graph. I would recommend to do individual western blots and include all data regardless that the number of replicates are different in the groups.

Response: The Western blots presented in figure 6 were obtained using a mixture (pool) of liver lysates derived from the respective group of mice. The relative intensities presented in the graphs are the average of three Western blots performed using these samples. This is now stated in the figure legend (p. 8, lines 225, 226). Only the MMP9 samples presented in S.Fig 2 were analyzed individually.

  • Figure legends 7 and 8 contain methodological explanations that are better suited in the materials and methods section.

Response: We have moved the information (that was not already presented in 'Methods')  to the 'Materials & Methods' section, as suggested by the reviewer.

  • There are 6 supplementary figures to the manuscript, but only two of them are referenced in the manuscript as suppl. figures 5 and 8 which doesn´t match the naming in the supplementary files.

Response: S.Fig.1 is referred to on page 4, line 142; S.Fig.2 is referred to on page 6, line 188; S.Figs. 3, 4. 5 are referred to on page 7, line 213, 216 and 217, respectively. S.Fig.6 is referred to on page 9, line 226.

5) The following typos are used consistently throughout the manuscript:

'oliver oil" instead of 'olive oil'

'CCL4' instead of 'CCl4'

'imag lab' instead of 'image lab'

Response: Many thanks for the comment; We have now corrected these typos.

Reviewer 2 Report

A well written MS describing higher Heparanase levels associated with damaged liver following CCl4 admission. 

A highly related MS was recently published and it is highly recommended to be cited, as it strengthens the results of the current MS, its details: 
J. Clin. Med. 2022, 11(6), 1672;&nbs

Author Response

Reviewer 2

A well written MS describing higher Heparanase levels associated with damaged liver following CCl4 admission. 

A highly related MS was recently published and it is highly recommended to be cited, as it strengthens the results of the current MS, its details: 
J. Clin. Med. 2022, 11(6), 1672;&nbs

Response: Thanks for the important information about the highly relevant and supportive recent study. The paper is now cited as Ref. # 22 (page 10, lines 276-278).

Reviewer 3 Report

Interesting work,

in the discussion, it is worth highlighting these  papers

Histochem Cell Biol. 2012 Feb;137(2):217-33.

Stem Cells Transl Med. 2019 Mar;8(3):271-284.  

Histochem Cell Biol  

2011 Mar;135(3):305-15.

 doi: 

Author Response

Reviewer 3

Suggestions for Authors

in the discussion, it is worth highlighting these  papers

Histochem Cell Biol. 2012 Feb;137(2):217-33.

Stem Cells Transl Med. 2019 Mar;8(3):271-284.  

Histochem Cell Biol  2011 Mar;135(3):305-15.

Response: We thank the reviewer for this important comment. The suggested publications are now cited as Ref. # 30, 31, and 32 (page 10, lines 332-336).